# The Toxic Effects and Mechanisms of Nano-Cu on the Spleen of Rats

**DOI:** 10.3390/ijms20061469

**Published:** 2019-03-22

**Authors:** Xuerong Zhou, Ling Zhao, Jie Luo, Huaqiao Tang, Min Xu, Yanyan Wang, Xiaoyu Yang, Helin Chen, Yinglun Li, Gang Ye, Fei Shi, Cheng Lv, Bo Jing

**Affiliations:** 1College of Veterinary Medicine, Sichuan Agricultural University, Chengdu 611130, China; xuerong_zhou@163.com (X.Z.); zhaoling010101@163.com (L.Z.); luojie010101@163.com (J.L.); turtletang@163.com (H.T.); xumin010101@163.com (M.X.); wangyanyan010101@163.com (Y.W.); yangxiaoyu010101@163.com (X.Y.); chenhelin0101@163.com (H.C.); yegang010101@163.com (G.Y.); shifei0202@163.com (F.S.); lvcheng010101@163.com (C.L.); jingbo020202@163.com (B.J.); 2School of Medicine, Tongren Polytechni College, Guizhou 554300, China

**Keywords:** nano-Cu, rats, spleen injury, immune system, oxidative stress

## Abstract

Research has shown that nano-copper (nano-Cu) can cause damage to the spleen and immune system yet their mechanisms of cytotoxicity are poorly understood. Our aim is to explore the potential immunotoxicity in the spleen of rats after nano-Cu exposure. The results of hematologic parameters, lymphocyte subsets, immunoglobulins, and histopathology indicated that copper obviously changed the immune function of the spleen. The levels of antioxidants (SOD, CAT, GSH-Px), oxidants (iNOS, NO, MDA), and anti-oxidative signalling pathway of Nrf2 (Nrf2 and HO-1) were strongly induced by nano-Cu. The expression of mRNA and protein of pro-/anti-inflammatory (IFN-γ, TNF-α, MIP-1α, MCP-1, MIF, IL-1/-2/-4/-6) cytokines were increased by nano-Cu. The expression of regulatory signal pathways, MAPKs and PI3-K/Akt were activated, which might be involved in the inflammatory responses and immunomodulatory processes of sub-acute nano-Cu exposure. The immune function of the spleen was repressed by nano-Cu induced oxidative stress and inflammation.

## 1. Introduction

Copper is an essential trace element in many metabolic processes. Copper plays an important role in maintaining the vitality of many enzymes, copper deficiency is associated with the progression of many diseases [1]. In livestock feed, copper was traditionally used as an additive for its growth promoting effects, but the overuse of copper in the care of livestock led to pollution of the environment [2,3]. Nano-Cu is another choice, because it is considered as being absorbed easily, and benefit to digestion of intestine, hematopoietic function, growth and immunity of animals at the lower doses [4,5]. Thus, nano-Cu is used more and more widely in animal feed, because it is more efficiency than normal Cu sources [6,7,8]. Although nano-Cu has been shown to have no teratogenic effects on chicken embryos, there are still concerns about its biosafety and potential to cause health defects in animals [9]. The United States Environmental Protection Agency has reported that artificial nanoparticles are classified as having unknown toxicity materials [10]. Nano-Cu is known to cause oxidative damage by inducing the production of reactive oxygen species in the organism [11]. Jani reported nanoparticles could reach lymphocytes nodes in the liver and spleen after crossing the gastrointestinal tract [12]. Once the nanoparticles are captured by splenic macrophages through endocytosis, they form cytotoxic aggregates [13,14].

The main function of the spleen is to clear the damaged, aging, and necrotic cells while also participating in the body’s immune response. The immune system is a sensitive target for toxicants [15]. In mice, low-dose toxicants are toxic for the immune system while being relatively safe for the other organs [16]. There is much evidence showing that nano-particles can interact with the immune system. The accumulation of nanomaterials in the tissue would create an imbalance of reactive oxygen species (ROS) homeostasis, with the liver and spleen acting as the main targets of this increased oxidative stress. Nano-Cu would accumulate in spleen and cause an unusual metal steady-state system, apoptosis, and immune dysfunction after intravenous injection [17]. TiO_2_ has a negative influence on the immunity of spleen in the mice by inducing the inflammatory and apoptotic cytokines [18]. Nano-Cu could cause a similar toxicity in the spleen [19], but as of yet the molecular mechanisms of nano-Cu toxicity has not been fully explained. Therefore, research must be conducted to elucidate the mechanism of nano-Cu induced toxicity on the spleen to ensure the safe and rational application of it in animal care.

To explore the effect of nano-Cu exposure on the immune system of rats, we examined the changes of immune cells, lymphocyte subsets, anti-bodies, and histopathology after 28 days of oral exposure. The present study was undertaken to comprehensively assess the immunomodulatory and inflammatory effect of sub-acute nano-Cu exposure in rats and further investigate the relevant mechanisms that mediate these effects.

## 2. Results

### 2.1. Morphology and Physicochemical Parameters of Nano-Cu

Figure 1A,B show the morphology of 1 μm Cu and 80–100 nm Cu characterized by SEM, respectively. Table 1 lists the physicochemical parameters of 1μm Cu and 80–100 nm Cu nano-particles by a Zeta sizer Nano ZS (Malvern Instruments, Malvern, UK). The results show that the particle size of 80–100 nm Cu and 1 μm Cu are mainly distributed in the range of 45–115 nm and 0.5–1.25 μm, respectively.

### 2.2. Concentration of Copper in the Spleen

The weight of rats in the micro-Cu (*p* < 0.05), nano-Cu medium- (*p* < 0.05), and high- (*p* < 0.01) groups are lower than control group (Figure 2A). In Figure 2B, copper is significantly accumulated in the spleen in a dose-dependent manner.

### 2.3. Nano-Cu Altered the Number of Blood Cells in Rats

Table 2 lists the hematologic results of the rats. WBC increases in the micro-Cu (*p* < 0.01), CuCl_2_ (*p* < 0.01), nano-Cu medium- (*p* < 0.05), and high- (*p* < 0.01) groups; lymphocytes increases in the micro-Cu (*p* < 0.01), CuCl_2_ (*p* < 0.01) and nano-Cu high- (*p* < 0.01) groups; RBC and HCT decrease in the mano-Cu medium- (*p* < 0.05) and high- (*p* < 0.01) groups, in a dose-dependent manner in the nano-Cu treated groups; PLT increases in the high-dose nano-Cu treated group (*p* < 0.05).

### 2.4. Nano-Cu Exposure Altered Lymphocyte Subpopulation in the Spleen

As shown in Figure 3, the lymphocyte subsets (CD3^+^, CD3^+^CD4^+^, CD3^+^CD8^+^, B, and NK cells) are decreased in the micro-Cu (*p* < 0.05), CuCl_2_ (*p* < 0.05), nano-Cu medium- (*p* < 0.05) and high- (*p* < 0.05) groups. With the increasing dosage of nano-Cu, the number of lymphocyte subsets decrease. Meanwhile, the ratios of CD3^+^CD4^+^/CD3^+^CD8^+^ are 5.84%, 16.63%, 13.44%, 18.39%, in the micro-Cu, CuCl_2_, nano-Cu medium- and high- groups, respectively, compared with the control group.

### 2.5. Nano-Cu Exposure Affected the Antibody Production

Figure 4 illustrates that the concentrations of IgA, IgG and IgM. IgA decreases in the micro-Cu (*p* < 0.01), CuCl_2_ (*p* < 0.01), nano-Cu medium- (*p* < 0.01) and high- (*p* < 0.01) groups; IgG decreases in the micro-Cu (*p* < 0.05), CuCl_2_ (*p* < 0.05), nano-Cu medium- (*p* < 0.05), and high- (*p* < 0.01) groups; IgM decreases in the micro-Cu (*p* < 0.05), CuCl_2_ (*p* < 0.01), nano-Cu medium- (*p* < 0.05) and high- (*p* < 0.01) groups dose groups. All the changes are in a dose-dependent manner in the nano-Cu treated groups.

### 2.6. Nano-Cu Exposure Induced Obvious Histopathology Changes

Figure 5 illustrates the histopathology of the spleens. There are no nano-Cu-related toxic histopathology changes in the control and micro-Cu groups (Figure 5A,B). In the CuCl_2_ group, the number of macrophages increase in the red pulp region (Figure 5C), denaturation of splenic trabecular arterial muscle cells, the inflammatory cell infiltration is also observed (Figure 5D). Macrophages increase in the nano-Cu low- (Figure 5E) and medium- dose (Figure 5F) groups. An amount of deposition of amyloid is exhibited in the nano-Cu high group (Figure 5G).

### 2.7. Analysis of the Oxidative Stress in the Spleen 

The results of oxidative stress are illustrated in Table 3. Compared with the control group, MDA increases in the micro-Cu (*p* < 0.05), nano-Cu medium- (*p* < 0.01), and high- (*p* < 0.01) groups. iNOS increases in the CuCl_2_ (*p* < 0.05), nano-Cu medium- (*p* < 0.05), and high- (*p* < 0.01) groups. NO, SOD, CAT and GSH-Px increase in the micro-Cu (*p* < 0.05), CuCl_2_ (*p* < 0.05), nano-Cu medium- (*p* < 0.05), and high- (*p* < 0.01) groups. nano-Cu induces obvious oxidative stress in the spleen in a dose-dependent manner.

### 2.8. Nano-Cu Exposure Induced Inflammatory Responses in the Spleen 

The Table 4 and Table 5 illustrate the results of gene transcription and protein expression. Compared with control group, the genes transcription for *TNF-α*, *MIP-1α*, *IL-1β*/*-2*/*-4*/*-6* are significantly increased in the micro-Cu (*p* < 0.05), CuCl_2_ (*p* < 0.05), nano-Cu medium- (*p* < 0.05), and high- (*p* < 0.01) groups. *IFN-γ*, *MCP-1* and *MIF* increase in the CuCl_2_ (*p* < 0.05), nano-Cu medium- (*p* < 0.05) and high- (*p* < 0.01) groups. The proteins levels of these inflammatory factors are consistent with increased gene expression.

### 2.9. Nano-Cu Exposure Activated MAPK, NrF2, and PI3K in the Spleen

Figure 6 and Figure 7 illustrate the levels of the gene transcription and proteins expression, respectively. In the Nrf2 signaling pathway, the genes transcription, total and phosphorylated proteins expression of Keap1 increase in the micro-Cu (*p* < 0.05), CuCl_2_ (*p* < 0.05), nano-Cu medium- (*p* < 0.05) and high- (*p* < 0.01) groups; *Bach1* gene transcription increases in the CuCl_2_ (*p* < 0.05), nano-Cu medium- (*p* < 0.05) and high- (*p* < 0.05) groups, and total proteins increases in the micro-Cu (*p* < 0.05), CuCl_2_ (*p* < 0.01), nano-Cu medium- (*p* < 0.05) and high- (*p* < 0.01) groups, while phosphorylated proteins expression increases in the CuCl_2_ (*p* < 0.05) and nano-Cu high (*p* < 0.05) groups; *Nrf2* and *HO-1* genes transcription, total and phosphorylated proteins expression increase in the micro-Cu (*p* < 0.01), CuCl_2_ (*p* < 0.01), nano-Cu low- (*p* < 0.05), medium- (*p* < 0.01) and high- (*p* < 0.01) groups. In the MAPKs signaling pathway, *JNK* gene transcription increases in the micro-Cu (*p* < 0.05), CuCl_2_ (*p* < 0.01) and nano-Cu high (*p* < 0.01) groups, and total and phosphorylated proteins increase in the micro-Cu (*p* < 0.01), CuCl_2_ (*p* < 0.01), nano-Cu medium- (*p* < 0.05), and high- (*p* < 0.01) groups; *p38* and *ERK_1/2_* genes transcription, total and phosphorylated proteins expression increase in the micro-Cu (*p* < 0.01), CuCl_2_ (*p* < 0.01), nano-Cu medium- (*p* < 0.05) and high- (*p* < 0.01) groups; *AP-1* gene transcription increases in the micro-Cu (*p* < 0.01), CuCl_2_ (*p* < 0.01), nano-Cu medium- (*p* < 0.01) and high- (*p* < 0.01) groups, and total and phosphorylated proteins expression increase in the micro-Cu (*p* < 0.01), CuCl_2_ (*p* < 0.01), nano-Cu low- (*p* < 0.05), medium- (*p* < 0.01) and high- (*p* < 0.01) groups; *CREB* and *COX-2* genes transcription, total and phosphorylated proteins expression increase in the micro-Cu (*p* < 0.05), CuCl_2_ (*p* < 0.01), nano-Cu low- (*p* < 0.05), medium- (*p* < 0.01) and high- (*p* < 0.01) groups. In the PI3K/Akt signaling pathway, *PI3K* gene transcription, total and phosphorylated proteins expression increase in the micro-Cu (*p* < 0.01), CuCl_2_ (*p* < 0.01), nano-Cu medium- (*p* < 0.05), and high- (*p* < 0.01) groups; *Akt* gene transcription increased in the CuCl_2_ (*p* < 0.01), nano-Cu medium- (*p* < 0.05) and high- (*p* < 0.01) groups, and total and phosphorylated proteins expression increase in the micro-Cu (*p* < 0.05), CuCl_2_ (*p* < 0.05), nano-Cu medium- (*p* < 0.01), and high- (*p* < 0.01) groups.

## 3. Discussion

Nano-Cu is a potential anti-bacterial and growth promoting material that can be used as an additive in animal feed. However, with the widespread use of nano-Cu, the risk of their unknown toxic side effects developing are becoming increasingly more likely. The spleen is one of the target organs of nano-Cu and its cytotoxic effects have produced serious injuries to this organ in vivo [20]. The spleen plays a crucial role in organizing the innate and adaptive immune response and is involved in the elimination of pathogenic microorganisms, immune response to vaccines, and immune surveillance [21]. NPs can communicate with various biological components of the immune system, triggering cell signaling cascades, and consequently cause unpredictable immune responses (activation or suppression) and even harmful outcomes. The goal of this study was to evaluate the cytotoxic impact of nano-Cu on the spleen of rats. nano-Cu caused changes in structure and function of the spleen, also oxidative stress and inflammation were observed that induced the activation of relevant regulated pathways.

The LD_50_ of 80 nm nano-Cu is 2075 mg/kg∙w, and it is considered as a moderately toxic substance [20]. The 200 and 400 mg/kg nano-Cu caused significant digestive function disorder and induced significant growth inhibition [21]. The intestinal epithelial cells were treated with different doses of nano-Cu (5, 10, 20 and 40 μg/mL) for 24–48 h and signs of cell injury and oxidative stress were observed [22]. The injured gastrointestinal wall would have impaired absorption efficiency and allow nano-Cu to easily penetrate the gastrointestinal wall [23]. After nano-Cu passed through the gastrointestinal wall, they would be captured by macrophages. These cells usually produce higher levels of ROS, causing an accumulation of oxidative glutathione (GSSG) to clear the intruder. Tarantino reported that ceruloplasmin, an enzyme synthesized in the liver, carries more than 95% of the total copper in healthy human plasma. Nano-cu can cause liver damage, so excessive ROS release may also be related to the dysfunction of ceruloplasmin [1]. These impacts further elicit inflammatory responses through distinct signaling pathways and cause cytokine secretion (e.g., interleukins (ILs), tumor necrosis factor (TNF-α)) with the changes of innate immunity and adaptive immunity [24]. In our results, nano-Cu increased the levels of WBC and expression of inflammatory cytokines, which means that the particles caused an obvious inflammatory response in the spleen in a dose-dependent manner. Micro-copper particles and Cu ions also induced an increase of inflammation cells and cytokines, but the degrees were lower than nano-Cu. Oxidative stress induced inflammation is one of the main cytotoxic effects induced by metal nano-particles [25]. 

Numerous experimental studies have supported the role of the immune system in responding to metallic nano-particles [26,27]. These particles can induce high levels of reactive oxygen species (ROS) upon interaction with cellular components and cause activation of pro-inflammatory signals leading to immune system imbalance [24,28]. T cells (CD3^+^) play a key role in adaptive immunity, which can be divided into Th cells (CD3^+^CD4^+^ T cells), and cytotoxic T cells (CD3^+^CD8^+^ T cells) based on their role in the immune response. Most studies have indicated that nano-particles have impacted T cell infiltration, expansion, activation, and their proliferation [29]. In our study, all three kinds of copper were shown to decrease the numbers of CD3^+^CD4^+^ and CD3^+^CD8^+^ cells. The number of CD3^+^CD4^+^ and CD3^+^CD8^+^ and the ratio of CD3^+^CD4^+^/ CD3^+^CD8^+^ proved that the imbalance of cellular immunity was caused by nano-Cu [30]. Nano-Cu has also impacted the numbers of B and NK cells, as the number of B cells decreased, the expression level of antibodies for IgA, IgG, and IgM were significantly decreased. Nano-Cu seriously interfered with cellular and humoral immune functions in rat spleens, which will also lead to a decreased disease resistance in animals. 

In order to understand the causes of immune disorders caused by nano-Cu, we have studied multiple immunoregulatory pathways. Keap1-Nrf2/ARE signaling pathway is one of the most important antioxidant pathways, Nrf2 is a key factor for resisting oxidative stress. The gene transcription and protein expression of Nrf2 and HO-1 were increased in splenic cells, which indicated that in order to respond to the oxidative stress induced by nano-Cu, spleen cells produce a large number of antioxidant substances to reduce the damage caused by ROS [31]. The increased oxidative stress would activate redox-sensitive transcription factors including nuclear factor kappa B (NF-kB) and activator protein-1 (AP-1) via stress kinases MAPK (ERK, p38, and JNK). This leads to increased expression of pro-inflammatory cytokines, as well as inflammatory target proteins matrix metalloproteinase-9 (MMP-9) and cyclooxygenase-2 (COX-2). It is reported that the MAPK pathway plays important roles in the regulation of several genes involved in immune and inflammatory response, through the regulation of transcription factors NF-κB and AP-1 [32]. We found that the phosphorylation of MAPKs and AP-1 were strongly induced by nano-Cu in a concentration-dependent manner in the spleen at the levels of mRNA and protein expression. Some researchers have shown that MAPKs could up-regulate the expression of AP-1 and CREB [33,34]. In addition, AP-1 and CREB play an important role in transcriptional regulation of COX-2, a key enzyme in metabolism of arachidonic acid into PG [35,36]. PG is a main mediator of inflammation as well as a regulator of immunity [37]. PI3K/Akt are important transmitters in inflammatory pathways. Studies have provided evidence that increased PI3K/Akt signaling deregulates the humoral immune response [38]. Our finding shown that the mRNA and protein levels of CERB and PI3K/Akt were strongly increased by nano-Cu, and led to changes in the cytokines and immune cells of the spleen. These signals were also impacted by micro-Cu and Cu ions, which indicated that the immunotoxic effect of nano-Cu may be caused by both copper particles and their ionized forms. These results indicated that Keap1-Nrf2/ARE, PI3K/Akt, and MAPK pathways might be involved in nano-Cu induced immune and inflammatory process through activation byoxidative stress. Therefore, further research is needed on the role of nano-Cu and Cu ions on affecting the immune function of the spleen.

In conclusion, our present study demonstrated that nano-Cu induced obvious spleen damage and the oxidative-inflammatory-immune changes are associated with upregulation of several pro-inflammatory responses, oxi/antioxidants, and modulation of subtypes of CD3^+^CD4+/CD3^+^CD8+ T-cell numbers in the spleen of rats. Next, we also found the activation of MAPK, Nrf2, and PI3K pathways might be involved in the inflammatory responses and immunomodulatory processes to sub-acute nano-Cu exposure. Nano-Cu is more immunotoxic than normal Cu sources, so it is not suitable as a long-term additive in animal feed.

## 4. Material and Methods

### 4.1. Characterization of Nano-Cu

CuCl_2_.2H_2_O, micro-Cu (1 μm), and nano-Cu (80–100 nm) were purchased from Aladdin Industrial Corporation (Shanghai. China). Hydroxypropyl methylcellulose (HPMC, analytical grade), the suspending vehicle for the copper particles, was bought from Shanghai Ryon Biological Technology Co. Ltd. (Shanghai, China). All other chemicals were analytical grade. In the experiments, the micro-Cu and nano-Cu powder were suspended in 1% *w*/*v* HPMC solution, stirred for 2–3 min and oscillated for 15–20 min until fully dispersed before use. 

Scanning electron microscope (SEM; Phenom ProX, Nani Scientific Instruments Corp., Shanghai, China) was used for characterizing the sizes of nano-Cu. The distribution of particle sizes was characterized by polydispersity index through dynamic light scattering studies performed with a Zeta sizer Nano ZS (Malvern Instruments, Malvern, UK).

### 4.2. Animals and Experimental Procedures

Male Sprague-Dawley rats (aged four weeks; 100–120 g) were purchased from Chengdu Dossy Biological Technology Corp., (Chengdu, China). Rats were fed in a specific pathogen-free environment. All experiments were conducted under the Guide for Care and Use of Laboratory Animals [39]. All experimental procedures involving animals were approved by Animal Ethical Committee of Sichuan Agricultural University (approval code: SCAU2017101501; approved on 15 October, 2017).

After one week of adaptation, sixty male Sprague Dawley rats were randomly divided into six groups: group I (control group, 1% HPMC); group II (1 μm Cu group, 200 mg/kg); group III (CuCl_2_ group, 200 mg/kg); group IV (nano-Cu low group, 50 mg/kg); group V (nano-Cu medium group, 100 mg/kg nano-Cu); group VI (nano-Cu high group, 200 mg/kg). Rats were treated with different copper treatments by intragastric administration at 09:00 every morning continuously for 28 days. During the experiment, the behavior, posture, salivation, pupil changes, muscle tremors, excrements and hair of rats were observed and recorded each day. At the end point of the treatment, all rats were weighed and anaesthetized with isoflurane. Blood was collected through cardiac puncture under anesthesia. Animals were sacrificed after blood collection and the spleens were promptly removed, weighed, and stored at −80 °C for future use. A small part of the spleen would be fixed in 10% buffered formalin for histological examination.

### 4.3. Copper Concentration in the Spleen 

The 0.2 g spleens were dissolved with concentrated nitric acid and H_2_O_2_ at high temperature. Then the residual nitric acid was volatilized until the solution became transparent. Next, 3 mL of 2% nitric acid was added into the solution. The inductively coupled plasma mass spectrometry (Agilent Technologies Corp., Beijing, China) was used for determination of the copper concentration in the spleen. 20 ng/mL germanium was used for the internal standard element, purchased from Sigma Company.

### 4.4. Hematologic Analysis

Blood samples were collected to measure hematologic parameters by blood cell analyzer (BS-180, Shenzhen Mindray Bio-Medical Electronics Corp., Shenzhen, China). Hematologic parameters include red blood cell count (RBC), white blood cell count (WBC), lymphocytes, platelets (PLT), hemoglobin (HGB), and red blood cell specific volume (HCT). 

### 4.5. Lymphocyte Subpopulation Analysis 

The 0.2 g spleens were washed with PBS solution and passed through a cell strainer, then the red blood cells were removed from the suspension with red blood cell lysis buffer. Suspensions were then centrifuged, and supernatants were removed. Next, cell sedimentations were suspended with PBS again, and the concentration of cells is about 10^6^/mL. The cell suspensions were stained with antibodies (anti- CD3^+^, anti-CD3^+^CD4^+^, anti-CD3^+^CD8^+^, anti-CD19, and anti-NK1.1, purchased from BD Biosciences, Shanghai, China). The cells (CD3^+^, CD3^+^CD4^+^, CD3^+^CD8^+^, B, and NKT cells) were analyzed by flow cytometry (BD FACSVerse, BD Biosciences, Shanghai, China).

### 4.6. Immunoglobulin Analysis

Serum samples were separated to measure immunoglobulin by ELISA kits, including IgA, IgG, and IgM (Nanjing Jiancheng Bioengineering Institute, Nanjing, China). 

### 4.7. Histological Examination

The fixed spleens were embedded in paraffin wax and sliced into 3 μm thickness, mounted on glass slides and stained with haematoxylin-eosin. Finally, histology was assessed by BA400Digital light microscope (Motic China Group Corp., LTD, Xiamen, Fujian, China).

### 4.8. Oxidative Stress Analysis

The 0.5 g spleen were homogenized with 4.5 mL ice-cold saline (0.86% NaCl *w*/*v*). Then, suspensions were centrifuged, the supernatants were collected to measure the levels of oxidative stress. The concentration of malondialdehyde (MDA), superoxide dismutase (SOD), nitric oxide (NO), inducible nitric oxide synthase (iNOS), catalase (CAT), and glutathione peroxidase (GSH-Px) were examined by commercial kits according to the manufacturer’s instructions (Nanjing Jiancheng Bioengineering Institute, Nanjing, China).

### 4.9. mRNA and Protein Expression of Cytokines

The levels of gene transcription and protein expression of cytokines in the spleen were measured by real-time quantitative reverse transcriptase polymerase chain reaction (RT-PCR) [40] and Luminex [41] (suspension array, liquid chip). RT-PCR was used for measuring the transcription of genes. Synthesized primers for real-time PCR were designed by Huada Gene (Shenzhen Huada Gene Research Institute, Shenzhen, China). The RT-PCR primer sequences are listed in Table 6. In the process of Luminex, the spleen tissues were grounded into fine powder with liquid nitrogen, and 20 mg powder was lysed by 500 µL lysate buffer. Then, the supernatants, which contain protein, were gained by ultrasound and centrifugation. Next, loading control was conducted with β-tubulin MAPmate kit. Carry out 9-plex Multi-Pathway Magnetic Bead Panel (Millipore #46-680MAG, Germany) to measure the concentration of phosphorylate protein and Millipore #46-681MAG to measure the concentration of total protein. The factors include transcription factor TNF-α (tumor necrosis factor alpha), INF-γ (interferon-gamma), MCP-1 (monocyte chemotactic protein-1), MIP-1α (macrophage inflammatory protein-1 alpha), MIF (macrophage migration inhibitory factor), IL-1β/-2/-4/-6 (interleukin-1 beta/-2/-4/-6).

### 4.10. mRNA and Protein Expression of Nrf2, PI3K/Akt, and MAPKs Signalling Pathways

The gene transcription and protein expression of Nrf2, PI3K/Akt and MAPKs signaling pathways were chosen to be tested. Its contain Keap1 (kelch-like ECH-associated protein 1), Nrf2 (nuclear factor erythroid-2p45-related factor 2), Bach1 (BTB and CNC homology 1), HO-1 (heme oxygenase1), PI3K (phosphatidylinositol 3-kinase, Akt, ERK_1/2_ (extracellular regulated protein kinases), JNK (c-Jun N-terminal kinase), p38 (p38 mitogen-activated protein kinase), CREB (cAMP-response element binding protein), AP-1 (activating protein-1), and COX-2 (cyclooxygenase 2). Synthesized primers were designed by Huada Gene, (Shenzhen Huada Gene Research Institute, Shenzhen, China), and listed in Table 6.

### 4.11. Statistical Analysis

The data of each group were processed by SPSS 21.0 (IBM Corp., Armonk, NY, USA), represented by mean ± SD and drawn by GraphPad Prism 7.0 (GraphPad Software Corp., La Jolla, CA, USA). ANOVA was used for significant analysis, and *p* < 0.05 was considered as statistically significant.

## Figures and Tables

**Figure 1 ijms-20-01469-f001:**
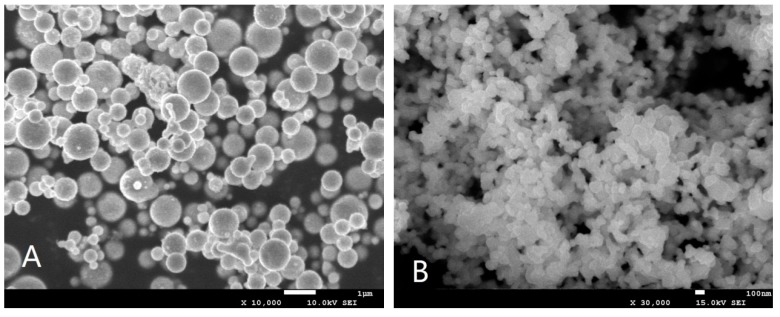
Surface characteristics of nano-copper: 1-µm copper particles (**A**) and Nano Cu particles (**B**). The average particle size of 1-µm copper particles and nano-copper particles is about 1 μm and 80–100 nm, respectively.

**Figure 2 ijms-20-01469-f002:**
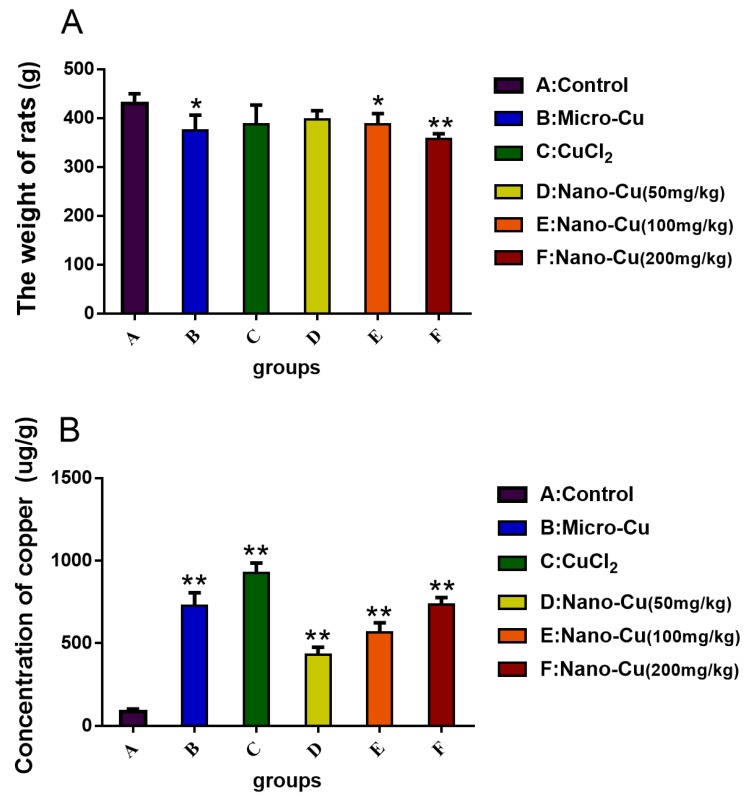
Weight of rats (**A**) and the concentration of copper ion (**B**) and in the spleen of each group of rats treated with oral administration of copper-containing preparations for 28 consecutive days. * *p* < 0.05 and ** *p* < 0.01 vs. control. Values represent means ± SD (*N* = 10). A: Control group, 1% *w*/*v* HPMC; B: micro-Cu group, 200 mg/kg BW; C: CuCl_2_ group, 200 mg/kg CuCl_2_ solution; D: nano-Cu low group, 50 mg/kg; E: nano-Cu medium group, 100 mg/kg; F: nano-Cu high group, 200 mg/kg.

**Figure 3 ijms-20-01469-f003:**
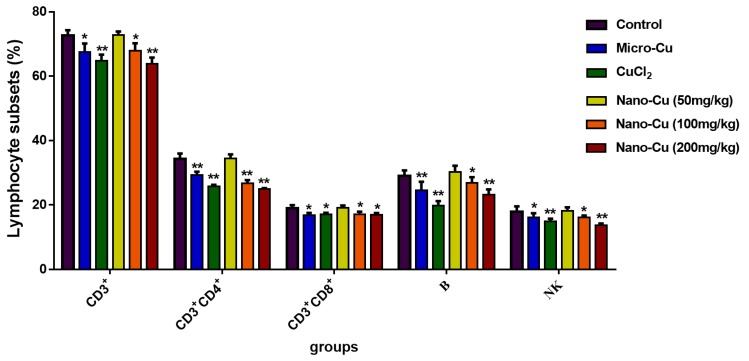
Lymphocyte subsets CD3, CD3CD4, CD3CD8, B, and NK cells in rats treated with oral administration for 28 consecutive days. * *p* < 0.05, and ** *p* < 0.01. Values represent means ± SD (*N* = 10).

**Figure 4 ijms-20-01469-f004:**
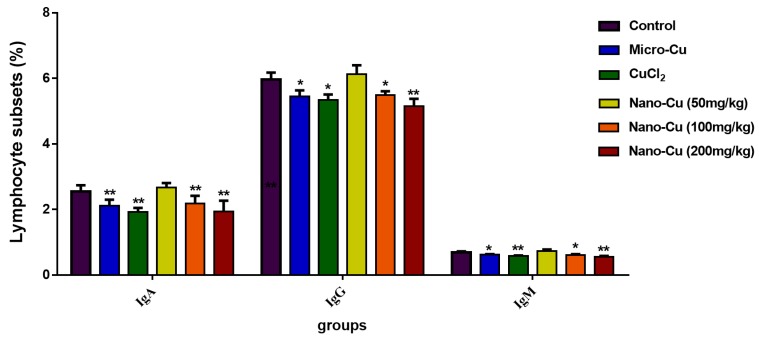
Immunoglobulin levels of IgA IgG, and IgM in rats treated with oral administration for 28 consecutive days. * *p* < 0.05 and ** *p* < 0.01. Values represent means ± SD (*N* = 10).

**Figure 5 ijms-20-01469-f005:**
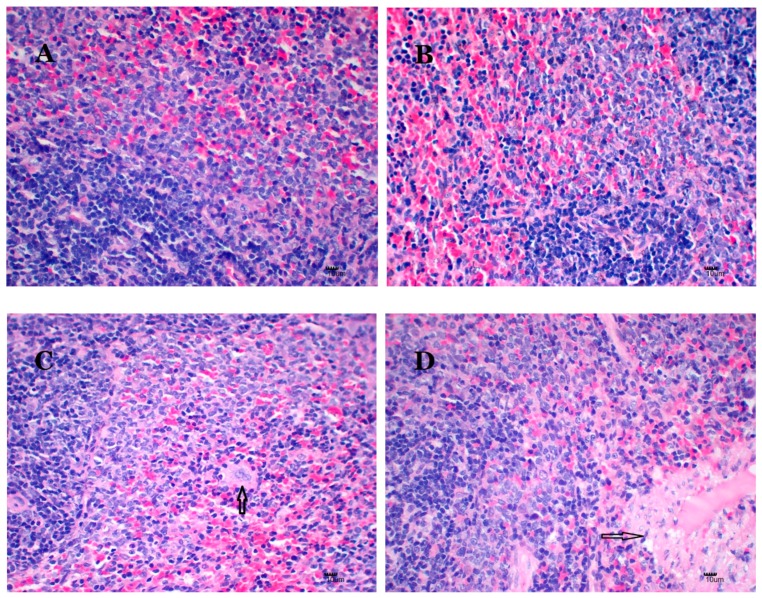
Hematoxylin-eosin staining of splenic sections of rats treated with oral administration of copper-containing preparations for 28 consecutive days. *N* = 3 per group, scale bar = 10 μm; 400×. (**A**) Control group, 1% *w*/*v* HPMC solution, or Group I with no histological abnormality; (**B**) micro-Cu group, 200 mg/kg BW, or Group II, without histological abnormality. (**C**) CuCl_2_ group, 200 mg/kg BW CuCl2 solution or Group III animals, with increased numbers of macrophages in the red pulp region (arrow); (**D**) degeneration of splenic trabecular artery muscle cells and infiltration of inflammatory cells (arrow); (**E**) nano-Cu low group, 50 mg/kg BW, or Group IV, with increased numbers of macrophages (arrow); (**F**) nano-Cu medium group, 100 mg/kg BW, or Group V, with increased numbers of macrophages (arrow); (**G**) nano-Cu high group, 200 mg/kg BW, or Group VI, with amyloid degeneration (arrow).

**Figure 6 ijms-20-01469-f006:**
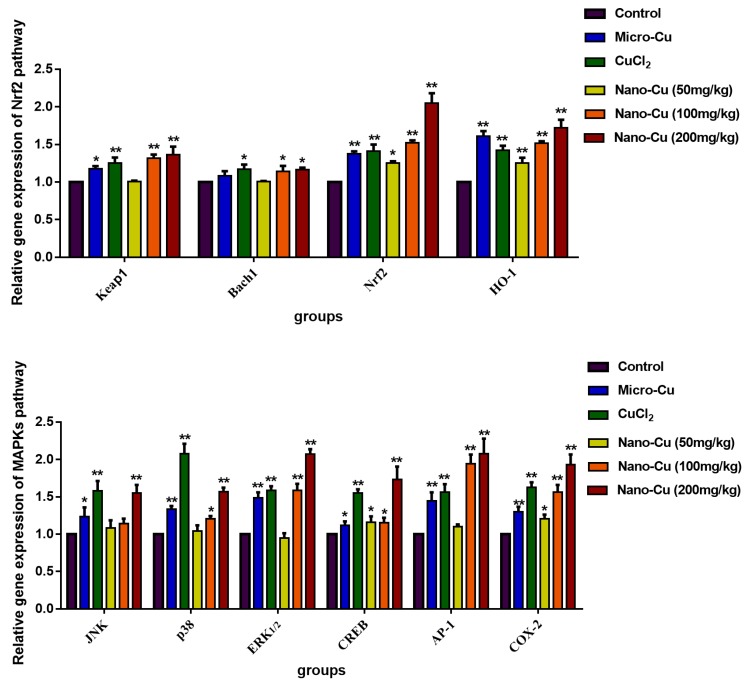
Expression levels of genes of the signaling pathway in rats given copper-containing preparations by oral administration for 28 consecutive days (2^−ΔΔCT^). * *p* < 0.05 and ** *p* < 0.01 vs. control. Values represent means ± SD (*N* = 10).

**Figure 7 ijms-20-01469-f007:**
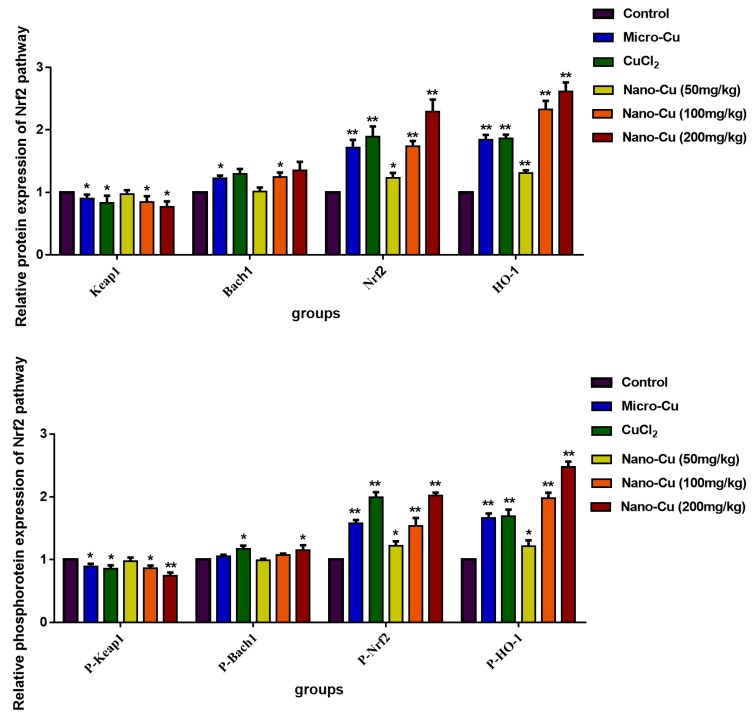
Expression levels of protein of the signaling pathway in rats given copper-containing preparations by oral administration for 28 consecutive days (test group/control). * *p* < 0.05 and ** *p* < 0.01 vs. control. Values represent means ± SD (*N* = 10).

**Table 1 ijms-20-01469-t001:** Physical and chemical parameters of 1 μm Cu and nano-Cu.

Particles	Average Size	Size Distribution	Purity (%)
Nano-Cu	80 nm	75 ± 35 nm	99.9
Micro-Cu	1 μm	0.85 ± 0.40 μm	99.9

**Table 2 ijms-20-01469-t002:** Hematologic parameters in rats treated with copper sources by oral administration over 28 consecutive days.

Index	Control	Micro-Cu	CuCl_2_·2H_2_O	Nano-Cu 80–100 nm		
	1%HPMC	200 (mg/kg)	200 (mg/kg)	50 (mg/kg)	100 (mg/kg)	200 (mg/kg)
RBC (10^12^/L)	7.8 ± 0.5	7.2 ± 0.4	7.2 ± 0.3	7.1 ± 0.3	6.9 ± 0.4 *	6.7 ± 0.5 *
WBC (10^9^/L)	9.2 ± 2.5	13.5 ± 2.6 **	17.1 ± 3.5 **	10.1 ± 2.5	10.9 ± 2.1 *	12.9 ± 2.3 **
lymphocytes (%)	8.8 ± 2.7	11.6 ± 3.4 **	12.7 ± 3.1 **	8.4 ± 1.6	9.5 ± 1.4	10.9 ± 2.3 **
PLT (10^9^/L)	1337.6 ± 100.5	1395 ± 104.6	1292.1 ± 135.4	1368.6 ± 100.6	1393.7 ± 65.1	1424.2 ± 129.9 *
HGB (g/dL)	159.3 ± 5.3	155 ± 8.1	155.1 ± 6.7	154.9 ± 9.9	156.7 ± 4.5	155.6 ± 12.3
HCT (%)	46 ± 2.8	42.8 ± 2.4	42.7 ± 2	48.6 ± 2.6	40.7 ± 1.3 *	40.9 ± 2.6 *

* *p* < 0.05 and ** *p* < 0.01 vs. Control. Values represent means ± SD (*N* = 10).

**Table 3 ijms-20-01469-t003:** Markers of oxidative stress in rats treated with copper sources by oral administration over 28 consecutive days.

Index	Control	Micro-Cu	CuCl_2_·2H_2_O	Nano-Cu 80–100 nm		
	1%HPMC	200 (mg/kg)	200 (mg/kg)	50 (mg/kg)	100 (mg/kg)	200 (mg/kg)
MDA (nmol/mgprot)	1.38 ± 0.07	1.55 ± 0.2 *	1.4 ± 0.04	1.31 ± 0.09	1.61 ± 0.11 **	1.75 ± 0.02 **
SOD (U/mgprot)	58.77 ± 6.35	67.46 ± 5.53 **	65.12 ± 3.51 *	57.38 ± 4.23	65.17 ± 5.39*	69.58 ± 6.59 **
NO (μmol/gprot)	0.135 ± 0.004	0.147 ± 0.014 *	0.155 ± 0.006 *	0.129 ± 0.008	0.149 ± 0.003**	0.154 ± 0.005 **
iNOS (U/mgprot)	0.55 ± 0.14	0.6 ± 0.03	0.67 ± 0.15 *	0.59 ± 0.02	0.69 ± 0.09 *	0.72 ± 0.11 **
CAT (U/mgprot)	10.8 ± 2.4	14.6 ± 2 **	14.2 ± 0.6 **	10.1 ± 2.3	15.9 ± 1.4 **	15.7 ± 0.8 **
GSH-Px (U/mgprot)	506.36 ± 41.07	613.74 ± 75.21 **	589.42 ± 16.08 *	545.04 ± 78.91	672.40 ± 42.94 **	751.45 ± 96.60 **

* *p* < 0.05 and ** *p* < 0.01 vs. Control. Values represent means ± SD (*N* = 10).

**Table 4 ijms-20-01469-t004:** Expression levels of genes associated with inflammation in rats treated with copper sources by oral administration over 28 consecutive days (2^−ΔΔCT^).

Gene	Control	Micro-Cu	CuCl_2_·2H_2_O	Nano-Cu 80–100 nm		
	1%HPMC	200 (mg/kg)	200 (mg/kg)	50 (mg/kg)	100 (mg/kg)	200 (mg/kg)
IFN-γ	1.00 ± 0.00	1.08 ± 0.07	1.19 ± 0.06 *	1.03 ± 0.01	1.17 ± 0.03 *	1.68 ± 0.08 **
TNF-α	1.00 ± 0.00	1.34 ± 0.10 **	1.45 ± 0.05 **	1.04 ± 0.01	1.43 ± 0.06 **	1.63 ± 0.08 **
MIP-1α	1.00 ± 0.00	1.37 ± 0.03 *	1.48 ± 0.04 **	1.01 ± 0.02	1.22 ± 0.02 *	1.74 ± 0.08 **
MCP-1	1.00 ± 0.00	1.05 ± 0.03	1.16 ± 0.03 *	1.05 ± 0.08	1.15 ± 0.08 *	1.44 ± 0.08 **
MIF	1.00 ± 0.00	1.09 ± 0.60	1.18 ± 0.20 *	1.06 ± 0.03	1.34 ± 0.50 **	1.57 ± 0.90 **
IL-1β	1.00 ± 0.00	1.31 ± 0.03 *	1.73 ± 0.02 **	1.00 ± 0.02	1.13 ± 0.04 *	1.54 ± 0.05 **
IL-2	1.00 ± 0.00	1.17 ± 0.01 *	1.23 ± 0.04 *	1.00 ± 0.01	1.18 ± 0.01 *	1.43 ± 0.03 **
IL-4	1.00 ± 0.00	1.16 ± 0.03 *	1.19 ± 0.05 *	1.08 ± 0.02	1.16 ± 0.02 *	1.51 ± 0.02 **
IL-6	1.00 ± 0.00	1.13 ± 0.05 *	1.17 ± 0.05 *	1.01 ± 0.02	1.22 ± 0.01 *	1.48 ± 0.03 **

* *p* < 0.05 and ** *p* < 0.01 vs. Control. Values represent means ± SD (*N* = 10).

**Table 5 ijms-20-01469-t005:** Expression levels of proteins associated with inflammation in rats treated with copper sources by oral administration over 28 consecutive days.

Protein (pg/mL)	Control	Micro-Cu	CuCl_2_·2H_2_O	Nano-Cu 80–100 nm		
	1%HPMC	200 (mg/kg)	200 (mg/kg)	50 (mg/kg)	100 (mg/kg)	200 (mg/kg)
IFN-γ	156.3 ± 5.7	165.0 ± 4.6 *	174.7 ± 8.5 *	159.3 ± 5.5	169.3 ± 11.5 *	252.0 ± 11.0 **
TNF-α	91.6 ± 2.2	153.1 ± 4.6 **	142.8 ± 5.0 **	91.7 ± 8.2	163.6 ± 4.7 **	205.3 ± 3.3 **
MIP-1α	19.4 ± 2.9	27.4 ± 4.7 *	28.9 ± 3.3 **	19.8 ± 1.9	23.6 ± 4.7 *	33.5 ± 5.6 **
MCP-1	103.8 ± 1.8	107.6 ± 1.7	111.9 ± 2.1	104.1 ± 2.1	109.2 ± 1.7	119.0 ± 2.0 *
MIF	36.5 ± 3.2	38.6 ± 2.5	42.5 ± 6.1 **	37.1 ± 1.8	39.5 ± 2.2 *	58.88 ± 1.4 **
IL-1β	149.5 ± 3.6	194.3 ± 2.5 **	254.9 ± 13.8 **	148.3 ± 3	171.8 ± 7.5 *	198.5 ± 19.1 **
IL-2	14.8 ± 2.4	16.5 ± 3.1 *	18.2 ± 2.1 *	14.5 ± 1.5	16.6 ± 1.7 *	18.4 ± 3.7 **
IL-4	26.0 ± 2.7	22.1 ± 3.7 **	27.5 ± 3.8	25.0 ± 2.7	27.7 ± 4.1	30.2 ± 3.7 **
IL-6	96.5 ± 8.2	90.6 ± 5.3 *	94.0 ± 6.7	96.0 ± 8.5	103.1 ± 7.9 *	113.1 ± 10.7 **

* *p* < 0.05 and ** *p* < 0.01 vs. Control. Values represent means ± SD (*N* = 10).

**Table 6 ijms-20-01469-t006:** Real-time PCR primer pairs. PCR primers used in the gene expression analysis.

Primer name	Primer Sequence (5′ to 3′)		Product Size (bp)
*GAPDH*	F: CCTTCCGTGTTCCTACCCC	R: GCCCAGGATGCCCTTTAGTG	131
*IFN-γ*	F: TTCGAGGTGAACAACCCACA	R: CACTCTCTACCCCAGAATCAGC	131
*TNF-α*	F: AAGGGAATTGTGGCTCTGGG	R: ACTTCAGCGTCTCGTGTGTT	83
*MIP-1α*	F: GCCTGAGATTAGAGGCAGCA	R: AGGTGGCAGGAATGTTCTGG	89
*MCP-1*	F: GGGCCTGTTGTTCACAGTTG	R: TGAGTAGCAGCAGGTGAGTG	88
*MIF*	F: GGCCTCACTTACCTGCACC	R: AACCATTTATTTCTCCCGACC	108
*IL-1β*	F: TTGAGTCTGCACAGTTCCCC	R: ATGTCCCGACCATTGCTGTT	91
*IL-2*	F: AACAAGTCTGGGGTTCTCGG	R: TGTTGTGAGCGTGGACTCAT	102
*IL-4*	F: AACAAGTCTGGGGTTCTCGG	R: TGTTGTGAGCGTGGACTCAT	102
*IL-6*	F: CTGGTCTTCTGGAGTTCCGTT	R: AGAGCATTGGAAGTTGGGGT	175
*Keap1*	F: CCAGGTACATAGGTCTGGCTG	R: CCAGTACGCCTCTAGCTGAA	72
*Bach1*	F: AAGCTGAGTTTGGAGGCAGA	R: GTGCAAACCCACAATGGACC	88
*Nrf2*	F: GGCTGTGTGTTCTGAGTATCG	R: TCCATGTCCGTTGTAAGCCA	86
*HO-1*	F: AGCACAGGGTGACAGAAGAG	R: AACTCTGTCTGTGAGGGACT	118
*PI3K*	F: GTGCCTTAGCTCTCTCTGCT	R: ACTGGGTTTCCTCATGGCTG	163
*ERK_1/2_*	F: TCCTTGGGAGGGAAGATACC	R: ATGACAATCCCGTAGCTCCA	101
*JNK*	F: TGATGACGCCTTACGTGGTA	R: GGCAAACCATTTCTCCCATA	120
*p38*	F: AGACGAATGGAAGAGCCTGA	R: GGGATGGACAGAACAGAAGC	109
*CREB*	F: GAGAAGCCGAGTGTTGGTGA	R: ACTCTGCTGGTTGTCTGCTC	176
*AP-1*	F: AAGTAGCCCCCAACCTCTCT	R: CACCCCAGCATACAGACACT	85
*COX-2*	F: TTTCAATGTGCAAGACCCGC	R: TACAGCTCAGTTGAACGCCT	120

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
