# Peer review of "The Toxic Effects and Mechanisms of Nano-Cu on the Spleen of Rats"

_ijms, 2019, doi:10.3390/ijms20061469_

Reviewer 1 Report

The authors have studied safety of nanoCu to be used in animal feed. They claim that nanoCu is much safer than bulk Cu currently being used. They have collected large amount of data ranging from absorption, hematology, inflammation, and gene and protein expression.

*I strongly suggest to delete the NFkB data or analyze samples using EMSA for NFkB  translocation in nucleus.

I strongly recommend that the be edited by someone with knowledge of working with scientific manuscripts. Authors have data, but they are not able to explain it. Entire manuscript must be revised. I have made some suggestions in the manuscript. 

Author Response

Dear Editor:

Thank you very much for your constructive advices on our manuscript. We have revised our manuscript in the past week. If there are still any problems with the manuscript, please let us know and we will be willing to revise the manuscript.

Review 1

1. Q: Editor strongly suggest to delete the NF-kB data or analyze samples using EMSA for NF-kB translocation in nucleus, and recommend that the be edited by someone with knowledge of working with scientific manuscripts. Authors have data, but they are not able to explain it. Entire manuscript must be revised. I have made some suggestions in the manuscript.

A: I have deleted or corrected some data. In addition, I have revised languages, formats and content of the manuscript. I have rewritten the abstract, introduce and discussion of manuscript.

Reviewer 2 Report

The authors studied toxicity of various formulations of Cu, which are used as livestock feed. Particularly, they focused on the nanoformulation of copper, which is believed to replace the inorganic copper. The authors convincingly presented various negative consequences of all copper sources on the immune system and spleen. However, what is missing is correlating all the tested doses on the efficacy of animal growth. If animal growth is satisfactory and copper accumulates in the spleen, then it might be that we should we be aware to not consume spleen from animals fed with copper, but still typically spleen is rather not used as a human food. It would be also interesting to know doses of copper that are present in the meat derived from copper fed animals. The manuscript would benefit with more advanced graphical presentation of statistical calculations. Are there any real health consequences of the observed abnormalities? Probably animals should be kept much longer to see it. The metabolism in rats can be radically different than in livestock animals, so some confirmation in large animals would add a value to the current study.

There are also multiple typographical errors (see below few examples), therefore the entire manuscript should be proofread carefully.

“Gonzaleseguia” “by intravenously” “with the give statement”

Author Response

Dear Editor:

Thank you very much for your constructive advices on our manuscript. We have revised our manuscript in the past week. If there are still any problems with the manuscript, please let us know and we will be willing to revise the manuscript.

Review 2

1. Q: what is missing is correlating all the tested doses on the efficacy of animal growth.

A: To correctly research the toxic effect of Nano-Cu on the spleen of rats, we design the dosage according to the LD50 which we have done before. Also, I have added the data of rat’s weight to exhibit the efficacy Nano-Cu on the animal’s growth.

2. Q: If animal growth is satisfactory and copper accumulates in the spleen, then it might be that we should we be aware to not consume spleen from animals fed with copper, but still typically spleen is rather not used as a human food.

A: Although spleen is not used as a human food, it’s very important for the growth and development of animals. Besides, spleen is the most important organ to filter blood, to monitor and remove damaged or abnormal cells and foreign antigens in the body. When spleen get injured, the blood system and immune system of body would be into destroyed. Other tissue would be attacked by foreign antigens without protection provided by the splenic immunity. Therefore, researching the toxicity Nano-Cu caused on the spleen of rats is an important stage to study the toxicity Nano-Cu caused on the heavy livestock as feed additive and to provide scientific basis of clinical application in the future. 

3. Q: It would be also interesting to know doses of copper that are present in the meat derived from copper fed animals.

A: It is also a very interesting research field to know the effect of Nano-Cu caused on the animal muscle, while the main theme of our experiment is known the effect of Nano-Cu cause on the spleen. Because spleen is the most important organ to filter blood, to monitor and remove damaged or abnormal cells and foreign antigens in the body. It is also a very interesting research field to know the effect of Nano-Cu caused on the animal muscle, while the main theme of our experiment is known the effect of Nano-Cu cause on the spleen. Because spleen is the most important organ to filter blood, to monitor and remove damaged or abnormal cells and foreign antigens in the body. Therefore, although spleen is not used as a human food, it’s very important for the growth and development of animals. Besides, at present, there are many researches about the toxic of Nano-Cu caused on the liver and kidney, but few about the spleen and immunity.

4. Q: The manuscript would benefit with more advanced graphical presentation of statistical calculations.

A: The graphical presentation of statistical calculations of manuscript have been advanced.

5. Q: The manuscript would benefit with more advanced graphical presentation of statistical calculations.

A: There are some real health consequences in the weight between Micro-Cu, Nano-Cu medium- and high- group, compared with control group.

6. Q: Probably animals should be kept much longer to see it.

A: Conduct the sub-chronic of experiments is aim to search for the dosage and measuring indexes of chronic experiments.

7. Q: The metabolism in rats can be radically different than in livestock animals, so some confirmation in large animals would add a value to the current study.

A: Researching the toxicity Nano-Cu caused on the spleen of rats is an important stage to study the toxicity Nano-Cu caused on the heavy livestock as feed additive. We could speculat the primary toxic effect of Nano-Cu cause on the heavy livestock, and make a perfect test scheme to research the molecular of injury of spleen.

8. Q: There are also multiple typographical errors, therefore the entire manuscript should be proofread carefully.

9. A: I have revised languages, formats and content of the manuscript. Besides, I have rewritten the abstract, introduce and discussion of manuscript.

Round  2

Reviewer 1 Report

Authors have addressed my concerns.

Author Response

Thank for your suggestion which could improve our manuscript.

Reviewer 2 Report

The authors improved manuscript according to my suggestions. While still language correction is needed as far as I know it can be performed by in-house English Editor.

Author Response

Thank for your suggestion which could improve our manuscript, and we do a language correction under the help of an American friend.